# Hot and Cold Cognitive Disturbances in Parkinson Patients Treated with DBS-STN: A Combined PET and Neuropsychological Study

**DOI:** 10.3390/brainsci12050654

**Published:** 2022-05-16

**Authors:** Louise M. Jørgensen, Tove Henriksen, Skirmante Mardosiene, Ottilia Wyon, Sune H. Keller, Bo Jespersen, Gitte M. Knudsen, Dea S. Stenbæk

**Affiliations:** 1Neurobiology Research Unit, Department of Neurology, Copenhagen University Hospital-Rigshospitalet, 2100 Copenhagen, Denmark; ottilia.wyon@dadlnet.dk (O.W.); gitte@nru.dk (G.M.K.); dea@nru.dk (D.S.S.); 2Copenhagen Spine Research Unit, Center for Rheumatology and Spine Disease, Copenhagen University Hospital-Rigshospitalet, 2600 Glostrup, Denmark; 3Faculty of Health and Medical Sciences, University of Copenhagen, 2200 Copenhagen, Denmark; 4Department of Neurology, Copenhagen University Hospital-Bispebjerg, 2400 Copenhagen, Denmark; tove.henriksen@regionh.dk (T.H.); skirmante.mardosiene@regionh.dk (S.M.); 5Department of Clinical Physiology and Nuclear Medicine, Copenhagen University Hospital-Rigshospitalet, 2100 Copenhagen, Denmark; sune.hoegild.keller@regionh.dk; 6Department of Neurosurgery, Copenhagen University Hospital-Rigshospitalet, 2100 Copenhagen, Denmark; bo.jespersen@regionh.dk; 7Department of Psychology, University of Copenhagen, 1353 Copenhagen, Denmark

**Keywords:** Parkinson, deep brain stimulation, subthalamic nucleus, non-motor symptoms, cognition, mood, serotonin (5-HT), neuroimaging, positron emission tomography

## Abstract

Patients with Parkinson’s disease (PD) often suffer from non-motor symptoms, which may be caused by serotonergic dysfunction. Deep Brain Stimulation (DBS) in the subthalamic nucleus (STN) may also influence non-motor symptoms. The aim of this study is to investigate how the cerebral 5-HT system associates to disturbances in cognition and mood in PD patients with DBS-STN turned on and off. We used psychological tests and questionnaires to evaluate cognitive function and the effects on mood from turning DBS-STN off. We applied a novel PET neuroimaging methodology to evaluate the integrity of the cerebral serotonin system. We measured 5-HT1BR binding in 13 DBS-STN-treated PD patients, at baseline and after turning DBS off. Thirteen age-matched volunteers served as controls. The measures for cognition and mood were correlated to the 5-HT1BR availability in temporal limbic cortex. 5-HT1BR binding was proportional to working memory performance and inverse proportional to affective bias for face recognition. When DBS is turned off, patients feel less vigorous; the higher the limbic and temporal 5-HT1BR binding, the more they are affected by DBS being turned off. Our study suggests that cerebral 5-HTR binding is associated with non-motor symptoms, and that preservation of serotonergic functions may be predictive of DBS-STN effects.

## 1. Introduction

Parkinson’s Disease (PD) is one of the most common movement disorders, affecting 1% of the population above 60 years of age [1] and causing both motor and non-motor symptoms. It is characterized by neuronal degeneration and progressive striatal loss of the dopamine (DA)-producing cells, leading to dopaminergic dysfunction and the characteristic features of bradykinesia and rigidity in PD [1]. Although focus has initially been on the degeneration of the dopaminergic system, increasing attention is now given to neuronal degeneration outside the nigrostriatal pathways and other brain monoamine systems, such as serotonin (5-HT) [2]. Accordingly, there is a growing interest in non-dopaminergic medications and search for neuroprotective agents [1].

As reviewed [3], there is evidence from both biochemical, imaging and post-mortem studies that 5-HT and its biomarkers are reduced in PD with a regional distribution distinct from that of DA. 5-HT is involved in regulation of several physiological functions, e.g., sleep, digestion, sexual function, mood and cognition [4], of which dysfunction is generally known as a non-motor symptom in PD [5,6]. Deficits in the 5-HT system may thus be an important element underlying the non-motor symptoms of PD, which constitute a huge burden on quality of life in patients with PD [7,8].

In recent decades, researchers have put forward a two-system model in higher human cognition describing the existence of two qualitatively different mental systems for processing information, one that is emotion-independent, rational, rule-based, deliberate and slow (cold cognition) and the other emotion-dependent, automated, associative, led by emotion and rapid (hot cognition) [9,10], also known as affective cognition. Thus, hot cognition refers the interface between emotion and cognition in humans; in other words, hot cognition includes the cognitive processing of emotional stimuli, unlike cold cognition, which does not include processing of emotional stimuli.

The affective bias in hot cognition describes the relationship between processing of negative and positive emotional input, where a negative bias is the tendency to register negative stimuli more readily [11,12]. While it is considered normal to have a positive affective bias in memory, it is widely accepted that depressives have a negative bias in memory [13], and it has also been observed in schizophrenia, anxiety, attention deficit hyperactive disorder (ADHD) and addiction [14]. Both hot and cold cognition are involved in executive functions and mood, commonly impaired in PD patients at all stages. While disruption of domains for cold cognition are well described in PD patients, and also in those treated with deep brain stimulation (DBS) [15], the domains for hot cognition remain unexplored. We have recently established and validated two instruments for hot cognition, which assess the affective bias, the EMOTICOM [16] test battery for recognition of facial expression and the Verbal Affective Memory Task [17] (VAMT-26) for affective words.

We have recently used a novel functional PET neuroimaging methodology to measure the preservation of serotonergic neurons and capacity to release serotonin in PD patients treated with DBS-STN [18]. With [^11^C]AZ10419369 PET, we demonstrated a significant loss of 5HT1BR availability in the frontal and parietal cortex, a deficit which correlated with the degree of motor dysfunction. When DBS-STN was turned off, the brain regions with the best preservation of presynaptic serotonin function, namely the temporal and limbic cortex, responded by releasing serotonin. This suggests that the presynaptic serotonergic terminals in the temporal and limbic cortex are still relatively preserved, whereas the frontoparietal regions are more affected and have lost their serotonin-releasing capacity. The temporal and limbic cortex are generally associated with important cognitive functions, such as understanding language, learning, remembering verbal information, facial emotional recognition and processing emotional input [19].

Deep brain stimulation (DBS) of the subthalamic nucleus (STN) constitutes a supplementary treatment option for alleviation of motor symptoms and pharmaceutical side effects in PD, although only applicable to a highly selected population of patients. A vast amount of research indicates that the 5-HT system is also involved in DBS treatment in PD, as DBS induces both temporary and stationary effects on mood and behavior [15,20,21]. While many PD patients experience affective side effects, such as hypomania and depression within three months after DBS surgery [20], there are also multiple case series suggesting that DBS may also alleviate non-motor symptoms, and there is level I evidence for the effect of DBS on improvement of anxiety [15]. Deficits in the static and dynamic integrity of the cerebral 5-HT system may contribute to the non-motor symptom severity of hot and cold cognition in PD patients treated with DBS-STN.

The aim of the present study was to exploit [^11^C]AZ10419369 PET as a proxy for non-motor symptoms in PD patients treated with DBS-STN by evaluating: (I) group differences in hot and cold cognition between DBS-STN-treated PD patients and age-matched healthy controls, (II) the association between hot and cold cognition and the integrity of the 5-HT system in the temporal and limbic cortex across all participants and (III) the effects on mood from turning DBS-STN off in patients with PD. First, we hypothesized to find a negative affective bias and reduced memory capacity in PD patients, when compared to controls. Second, we hypothesized the reduction in cognitive function is proportional to the [^11^C]AZ10419369 PET non-displaceable binding potential at baseline (BP_ND_0) and 5-HT-releasing capacity in the temporal and limbic cortex. Third, we hypothesized that turning off DBS-STN induces worsening in mood state proportional to the BP_ND_0 and 5-HT-releasing capacity in the temporal and limbic cortex.

## 2. Materials and Methods

### 2.1. Subjects

We included 13 [age 60 ± 7] non-depressed and non-demented PD patients treated with DBS-STN from the movement disorder clinic at University Hospital of Copenhagen, and 13 age-matched healthy volunteers served as controls (HC). Details of the study population, recruitment, in-and exclusion criteria, demographics and clinical characteristics have previously been reported [16]. Missing data from the participants are specified in Table 1 (n) and were due to the following reasons: (a) one patient only underwent a PET scan, (b) one HC had to discontinue the study for reasons of discomfort while being placed in the PET scanner, and (c) eight HCs entered from other studies [16,22,23] without a complete dataset.

### 2.2. Study Design

Patients were admitted to the Department of Neurology and scanned the following day. On the day of admission, the presence of non-depression and non-dementia was verified and three cognitive tests for cold and hot cognition were applied. The following day, patients underwent a PET scan, where DBS was turned off 45 min after the start of the scan. Symptom severity of PD and mood state was assessed immediately before the start of the PET scan and once again after the scan, where DBS was turned off. Baseline measures constitute data obtained before turning DBS-STN off. After the PET scan, upon return to the ward, all patients reported themselves in habitual condition and were discharged either immediately or the day after, according to their own wish. HC were not hospitalized. They underwent the study elements orderly in a single day and without DBS-STN.

### 2.3. Primary Outcomes

#### 2.3.1. Cognitive Measures

The Letter-Number Sequencing [24] (LNS) from the Wechsler Adult Intelligence Scale (WAIS-III) was used to examine verbal working memory capacity and executive functions. It is an auditive test of 10 min, in which the tester reads a combination of jumbled letters and numbers with increasing difficulty (e.g., 2-C-A-3-B-1) and the participant is asked to recall the numbers first in ascending order and then the letters in alphabetical order (e.g., 1-2-3-A-B-C). It contains seven sets of letters and numbers and each set contains three trials. The test is stopped after three incorrect trials within the same set of letters and numbers. The number of correct trials is the main outcome of the test, and a higher score indicates a better working memory.

The Verbal Affective Memory Test (VAMT-26) [17] is a computerized test used to assess three subclasses of immediate (IMR) short-term (STM) and delayed (LTM) verbal memory recall. Each subclass can be described by a total score and three categories of words: positive, negative and neutral. The words are all commonly encountered and monosyllabic [17]. The recall memory outcomes are strongly correlated, and here, a weighted average for IMR, STM and LTM is applied. The affective bias (VAMT-26 bias) is calculated by subtracting the weighted average of negative words recalled from the weighted average of positive words recalled.

The EMOTICOM [16] test battery for emotional processing of Emotion Recognition (ER) is a computerized test which assesses the hit rate of trials in which a given facial expression is correctly identified. The facial expressions are positive (happy) and negative (anger/fear/sad). The primary outcome here in sight is the affective bias for ER (ER bias), which is calculated as the difference between scores for positive and negative facial expressions [16].

#### 2.3.2. Mood State Measures

The Profile of Mood States (POMS) [25] is a self-reported rating scale used to assess transient, distinct mood states described by a total score of mood disturbance (TMD) and six factors: Tension or Anxiety (T), Depression or Dejection (D), Anger or Hostility (A), Vigor or Activity (V), Fatigue or Inertia (F) and Confusion or Bewilderment (C). Participants were examined at baseline and patients were examined once more after the PET scan, where DBS was turned off to allow for measurement of change in mood. The effect on transient mood state from turning DBS off was described by the difference in TMD scores (ΔTMD) between the two conditions, DBS-OFF minus DBS-ON.

#### 2.3.3. Brain Imaging with PET [^11^C]AZ10419369

The radiochemical production of [^11^C]AZ10419369 was done as previously reported [26]. PET scanning was conducted with a high-resolution research tomography (HRRT) PET scanner (CTI/Siemens, Knoxville, TN, USA) and co-registered to an MRI brain scan. The MRI and PET scanning protocols, processing and quantification of [^11^C]AZ10419369 binding has previously been detailed [18]. Non-displaceable binding potentials (BP_ND_s) were computed by the Extended Simplified Reference Tissue Model [27], with the cerebellum as a reference. The PET data at baseline (BP_ND_0) and when DBS-STN was turned off (BP_ND_1) have previously been reported [18] and tested for group differences between patients and healthy controls.

### 2.4. Statistical Analyses

The sample size was based on a previous test-retest study in pigs with the PET radioligand [^11^C]AZ10419369 [28], where 11 PD patients is the estimated sample size required to demonstrate a 4% change in cortical binding after turning DBS-STN off. The estimate is given a significance level of 0.05, power of 0.8 and an effect size determined by the BP_ND_0 (mean ± SD) at baseline (0.71 ± 0.10) and BP_ND_1 after intervention with escitalopram (0.68 ± 0.12) in a therapeutically relevant dose (0.28 mg/kg). The Interclass Correlation Coefficient between groups was 0.98, as previously detailed [18].

Hypothesis I was tested for group differences, patients and controls, in affective bias (VAMT-26 bias and ER bias) and working memory capacity (LNS) at baseline using the unpaired *t*-test after verification of normality with Q-Q plots and Kolmogorov–Smirnov test. Hypothesis II was tested individually for each cognitive measure (LNS, VAMT-bias and ER-bias) across all participants in each region of interest (limbic and temporal cortex) for the association between [^11^C]AZ10419369 PET and cognitive outcome by multiple linear regression with either one of the primary outcomes described for each cognitive measure as the dependent variable and the following predictors: BP_ND_0, age and L-DOPA equivalents. Hypothesis III was evaluated in patients for ΔPOMS-TMD between the two conditions (on and off), using paired *t*-test. Drivers of ΔPOMS-TMD were identified in a post hoc analysis of the six subscales composing POMS-TMD, tested individually between conditions using a paired *t*-test and correction for multiple comparisons, and the association with [^11^C]AZ10419369 PET was tested individually for each region of interest (limbic and temporal cortex) by multiple linear regression with the subscale driver for ΔPOMS-TMD as the dependent variables and the following predictors: age and either BP_ND_0 or change in binding potential from turning DBS-STN off (ΔBP_ND_(%)), defined as (BP_ND_0−BP_ND_1)/BP_ND_0 × 100. The significance level was set at a *p*-value of 0.05 after family-wise correction for each hypothesis with the Bonferroni–Holm method.

## 3. Results

### 3.1. Baseline Data

The cognitive outcomes and mood state measurements at baseline are given in Table 1. The BP_ND_s of the limbic and temporal cortex have previously been reported [18].

#### 3.1.1. Cold and Hot Cognition

Patients show a significant reduction in LNS performance (Heges’ g: 2.4; mean LNS difference [95% CI]: −3.0 [−5.0;1.0]; punc: 0.005, pFWER = 0.02). We did not observe a significant reduction in the normative positive bias for VAMT-26 bias (punc = 0.94) and ER bias (punc = 0.16).

#### 3.1.2. Mood States

There was no group difference in POMS-TMD between patients and controls (punc = 0.94). None of the participants met the criteria for major depression [18].

### 3.2. Correlations between Cognition and [^11^C]AZ10419369 PET

The association between cognitive measures and BP_ND_ was similar in patients and HC. Therefore, the correlation analyses were performed across all participants to increase power, as the model only reached sub-significancy when analyzed in separate groups.

#### 3.2.1. Cold Cognition Correlates with BP_ND_0

Within our model, working memory performance is proportional with 5-HT1BR binding across patients and HC at baseline, as indexed by a significant correlation between the LNS score and [^11^C]AZ10419369 BP_ND_0 in both regions of interest: temporal cortex (Estimate, 6.0 [1.5 to 10.4], R^2^ = 0.58, punc = 0.002, pFWER = 0.01; units: LNS score per increase in unit temporal cortical BP_ND_0) and limbic cortex (Estimate, 4.6 [1.5 to 7.6], R^2^ = 0.61, punc = 0.001, pFWER = 0.006; units: LNS per increase in unit limbic cortical BP_ND_0), meaning that high 5-HT1BR binding was associated with a higher working memory performance (Figure 1a,b).

#### 3.2.2. Hot Cognition Correlates with BP_ND_0

Within our model, hot cognition is inversely proportional to 5-HT1BR binding at baseline, as indexed by a significant inverse correlation between ER bias and [^11^C]AZ10419369 BP_ND_0 in both regions of interest: temporal cortex (Estimate, −0.74 [−1.2 to −0.3], R^2^ = 0.40, punc = 0.04, pFWER = 0.16; units: ER bias per unit temporal cortical BP_ND_0) and limbic cortex (Estimate, −0.48 [−0.8 to −0.1], R^2^ = 0.45, punc = 0.08, pFWER = 0.24; units: ER bias per unit limbic cortical BP_ND_0), meaning that low 5-HT1BR binding was associated with a more positive bias (Figure 1c,d). The [^11^C]AZ10419369 BP_ND_0 did not correlate with VAMT-26 bias.

### 3.3. Turning DBS-STN Off

#### 3.3.1. Effects on Mood States

When DBS-STN was turned off in PD patients, there was a significant worsening in mood state, as indexed by an increase in POMS-TMD score (Hedges’ g: 21.1; mean increase (TMD score) [95% CI]: 17.3 [4.4 to 30.3]; *p* = 0.01).

In order to explore whether the observed worsening in mood state from turning DBS-STN off was constrained to one or more dimensions of POMS, we conducted a post hoc analysis of the six factors composing the POMS-TMD score. The post hoc analysis shows that worsening in POMS-TMD score, when DBS-STN was turned off, was driven by the POMS dimension for Vigor or Activity (POMS-V), as indexed by a decrease from baseline level of 21.8 ± 5.3 (Heges’ g: 7.0; mean decrease (POMS-V score) [95% CI]: −8.0 [3.7 to 12.3]; punc: 0.002, pFWER = 0.01).

#### 3.3.2. Worsening in Mood State Correlates with [^11^C]AZ10419369 PET

Within our model, worsening in Vigor and Activity is proportional to 5-HT1BR binding at baseline (Figure 2a,b), as indexed with ΔPOMS-V (off-on) and [^11^C]AZ10419369 BP_ND_0 in: temporal cortex (Estimate, 32.5 [3.6 to 61.3], R^2^ = 0.4, punc = 0.8, pFWER = 0.16; units: POMS-V score per unit temporal cortical BP_ND_0) and limbic cortex (Estimate, 25.3 [8.4 to 42.2], R^2^ = 0.48, punc = 0.02, pFWER = 0.04; units: POMS-V score per unit limbic cortical BP_ND_0), meaning that a higher 5-HT1BR availability was associated with more worsening in vigor and activity, when DBS is turned off.

The correlation between [^11^C]AZ10419369 BP_ND_0 and change in overall mood state (ΔPOMS-TMD) from turning DBS-STN off only reached borderline significance (*p* = 0.10) in the limbic cortex. We did not observe an association between the ΔPOMS-V and the regional change in BP_ND_ between the two conditions, DBS turned on and off.

## 4. Discussion

First, we demonstrate a lower working memory capacity in PD patients treated with DBS-STN, when compared to age-matched controls, but not in affective bias of processing emotional salient information. Second, we show that the 5-HT1BR availability in temporal and limbic cortex is proportional with the working memory capacity, and inverse proportional with the affective bias for recognition of emotional face expression across patients and controls. Finally, when DBS-STN is turned off, patients with higher limbic and temporal 5-HT1BR availability at baseline report more worsening in activity and vigor, whereas patients with low 5-HT1BR availability were less affected by DBS being turned off.

### 4.1. Baseline Cognition

When conducting a rigorous statistical analysis with conservative corrections for multiple comparisons, we observed a significantly lower working memory capacity and executive function in PD patients treated with DBS-STN compared to age-matched controls, as indexed by the LNS score. Deficits in executive functions in PD is well recognized, and commonly appear even prior to development of the motor symptoms leading to the diagnosis [29].

As reviewed [20], it has been suggested that functioning of some cognitive domains improve from DBS-STN, e.g., working memory and visuomotor sequencing, whereas functioning of other domains decline, most consistently reported for word fluency and also to some extent for verbal memory. The role of DBS-STN in the expression of cognitive changes remains unclear, and findings must be interpreted with caution due to small sample sizes and lack of a control group in most studies.

We did not demonstrate a difference in affective bias between groups of non-depressed PD patients treated with DBS-STN and HC, when measured by recognition of emotional face expression and verbal affective memory. Whereas a positive bias is considered normal [30], a negative bias is associated with a variety of other neuropsychiatric disorders, such as schizophrenia, depression, anxiety, ADHD and substance abuse [14]. Importantly, all our PD patients were non-depressed, which may explain why we did not measure a group difference in affective bias. A review [31] of face recognition in PD patients showed diverging outcomes, meaning that some studies have demonstrated a reduced capacity for face recognition, particular for face expression of fear and disgust, while others have not. In DBS-STN-treated PD patients, one study [32] reported a DBS-induced decline in angry face recognition, while another study [33] reported impairment in recognition of face expression for anger, sadness and disgust three months after surgery, when compared to presurgical hit rates. Such decline to decode negative emotional face expression will only cause a positive bias, such as observed in the present study.

### 4.2. Association to [^11^C]AZ10419369 PET

The data show that 5-HT1BR binding with [^11^C]AZ10419369 PET is associated with non-motor symptoms for hot and cold cognition, and that hot and cold cognition is anticorrelated. BP_ND_0 is proportional to working memory capacity and executive function, and inversely proportional to a loss of positive affective bias. We have previously reported that [^11^C]AZ10419369 PET is a proxy of increase in motor symptoms in PD [18]. Accordingly, patients with low BP_ND_0 have more motor symptoms and lower working memory performance, but they present with a more positive bias for emotional processing. By contrast, patients with high BP_ND_0 have none or little impairment in motor symptoms and executive function, but they exhibit a loss of normative positive bias for emotional processing.

This ambiguity of the affective bias and motor symptom severity showing opposite directions may be a sign of neurodegeneration and disease progression in PD, when interpreted in light of the so-called positivity effect, well-recognized in elder individuals. An age-related decline in decoding emotional face expressions has been repeatedly reported [34], with an overall larger decrease in capacity to decode negative face expression than positive face expressions, particularly sad and fearful facial expression, and sometimes also angry faces [35], leading to a more positive bias. This phenomenon, known as the age-related positivity effect, was also present in our control population, when comparing negative facial expressions of sad and fear to normative values obtained from a large population of younger controls [17,36].

We cannot conclude to which extent the decline in affective bias was caused by neurodegeneration itself from disease progression and age or was also influenced by DBS-STN or long-term dopamine replacement therapy. There is scientific support that dopamine transmission regulates processing of emotional information in Parkinson’s disease, as abnormalities in emotional processing respond to dopaminergic replacement therapy, and that this process is dissociated from, e.g., motor symptoms and disease duration [31,37]. However, L-DOPA equivalents were included as a predictor in our model.

A [^11^C]AZ10419369 PET study in non-depressed patients with PD not treated with DBS-STN did not demonstrate a significant correlation between 5-HT1BR binding and either non-motor and motor symptom severity or staging of the disease, which the authors ascribe to a lack of power or due to medication effects [38]. Here, we assess PD patients treated with DBS-STN and with a higher average score for symptom severity and disease staging, all of which may contribute to the significant association to [^11^C]AZ10419369 BP_ND_0 observed in our study.

### 4.3. Turning DBS-STN Off

When DBS was turned off, patients reported feeling less active and vigorous; the higher the limbic and temporal 5-HT1BR binding at baseline, the more they were affected by DBS being turned off. This suggests that PD patients with better preservation of presynaptic 5-HT function benefit more from DBS-STN because they report feeling less vigor after switching the DBS off, as compared to patients with lesser preservation of presynaptic 5-HT function. This interpretation would, however, need to be tested in an independent behavioral study. Secondary changes in the dopamine system, when turning DBS off, may possibly also affect vigor.

We have previously reported that higher 5-HT1BR availability is associated with preservation of the static and dynamic integrity of the 5-HT system, and is characteristic of patients with less motor symptom severity of PD at baseline and after turning DBS off. Here, we show that higher 5-HT1BR binding is also associated with the most worsening in mood state. These observations are in line with other studies describing separate mechanisms of cognitive and physiological fatigability in PD [39].

We suggest that patients with best preservation of the 5-HT system have greater capacity to process emotional salient information, meaning that they can register a lesser impairment when DBS is turned off and produce a right and proper emotional response.

### 4.4. Limitations

The findings should be interpreted with cautions as the study may not have sufficient power to accurately detect group differences in cognitive measures and correlations to [^11^C]AZ10419369 PET. Moreover, eight of the thirteen controls entered from other studies, with missing data from either one of the cognitive measures or PET scan, resulting in a lower sample size in the control group. While a sample size of thirteen is common for PET studies, this is relatively small for psychological studies, and we suggest these data should be replicated in an independent behavioral study, which could also include a non-DBS-STN group to study the effects of DBS-STN on mood and cognition in patients with PD.

Generalizability of the obtained results to PD patients at large may not apply, as one of the exclusion criteria in our study was psychiatric illness, e.g., depression, which is common in PD patients.

Since our study for logistical reasons only involved turning off DBS-STN, we cannot know if the reverse actions occur when turning DBS-STN on.

The EMOTICOM test battery uses a touch screen and is, thus, dependent on hand and arm movement, which is a concern when applying the test to patients with a hypokinetic disorder. However, the outcome measure for affective bias is relative, and thus, eliminates potential bias from motor inhibition and use of reaction time in the model.

## 5. Conclusions

We demonstrate that [^11^C]AZ10419369 PET is associated with hot and cold cognition across patients with PD treated with DBS-STN and healthy controls. Patients with the best integrity of the 5-HT system and least motor symptoms report less vigor when DBS-STN is turned off. We suggest that [^11^C]AZ10419369 PET may act as a predictor of the effect from surgery with DBS-STN on non-motor symptoms. More studies are needed to investigate the beneficial effects of [^11^C]AZ10419369 PET in the clinical work up of patients considered for this type of surgery.

## Figures and Tables

**Figure 1 brainsci-12-00654-f001:**
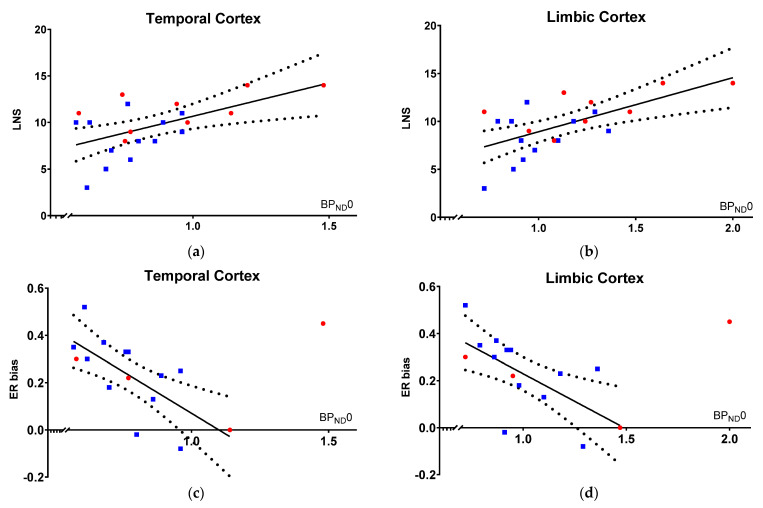
5-HT1BR PET binding vs. cold and hot cognition at baseline. [^11^C]AZ10419369 BP_ND_0 in PD patients (blue squares) and controls (red circles) versus: (**a**,**b**) LNS score in the temporal and limbic cortex; (**c**,**d**) ER bias in the temporal and limbic cortex. One control, with ER bias as outlier (>2SD), was excluded from the regression analyses. LNS and ERbias are anticorrelated at a sub-significant level (Spearman correlation coefficient −0.29, *p* = 0.27).

**Figure 2 brainsci-12-00654-f002:**
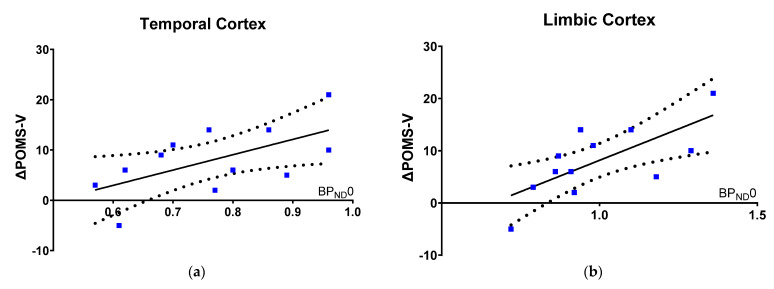
5-HT1BR binding vs. worsening in mood with DBS turned off. The association between change in POMS V (off-on), when DBS is turned off in PD patients, and (**a**) temporal and (**b**) limbic [^11^C]AZ10419369 BP_ND_0.

**Table 1 brainsci-12-00654-t001:** Descriptive information of cognition, mood and 5-HT1BR binding.

	Healthy Controls				Parkinson’s Disease			
	Mean	SD	Range	n	Mean	SD	Range	n
**Cold Cognition**								
Working memory (LNS)	11.3	1.9	[8–14]	11	8.3 *	2.6	[3–12]	12
**Hot Cognition**								
Verbal affective memory (VAMT-26 bias)	0.2	1.1	[−1.9–2.5]	11	−0.5	1.2	[−2.6–1.1]	12
Face recognition (ER bias)	23.4	16.4	[0–45]	5	24.1	16.9	[−8–52]	12
**Mood state**								
Profile of Mood State (POMS-TMD)	−6.5	14.6	[−19–21]	11	−6.5	15.0	[−20–24]	12
**[^11^C]AZ10419369 PET** ^(a)^								
Temporal cortex (BP_ND_0)	0.95	0.25	[0.58–1.48]	10	0.79	0.15	[0.72–1.39]	13
Limbic cortex (BP_ND_0)	1.28	0.32	[0.72–2.00]	10	1.02	0.21	[0.57–1.11]	13
Temporal cortex (ΔBP_ND_ (%))	3	11	[−19–20]	10	−11	9	[−29–1]	13
Limbic cortex (ΔBP_ND_ (%))	2	7	[−17–12]	10	−9	12	[−27–9]	13

Cognitive measures of working memory and affective bias at baseline with DBS turned on in healthy controls and patients with Parkinson’s disease treated with DBS-STN. Measures are: Letter Number Sequencing (n) for working memory, Verbal Affective Memory Test-26 for verbal affective memory, EMOTICOM face version for accuracy for decode of emotional face expression, and Profile of Mood State (POMS) total mood disturbance (TMD). Mean (average), Standard deviation (SD), Range [minimum and maximum scores] and number of participants (n). ER = correct hit rate of face recognition (%), Bias = positive score−negative score. * Significant reduction in LNS score using unpaired *t*-test (*p* = 0.02, after correction for multiple comparisons). ^(a)^ The PET data have been previously reported [17].

## Data Availability

Data are available upon request to the corresponding author.

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
