# Peer review of "Hot and Cold Cognitive Disturbances in Parkinson Patients Treated with DBS-STN: A Combined PET and Neuropsychological Study"

_brainsci, 2022, doi:10.3390/brainsci12050654_

Round 1
Reviewer 1 Report
At the manuscript " Hot and Cold Cognitive Disturbances in Parkinson Patients 2 Treated with Dbs-Stn: A Combined Pet and Neuropsychologi-3 Cal Study" by Dr. Louise M Jørgensen et al authors described results of investigation how the brain serotonin system associates to disturbances in cognitive functions and mood in Parkinson’ disease (PD) patients after deep brain stimulation (DBS). To evaluate cognitive function it was used psychological tests. Authors also used positron –emission tomography (PET) to evaluate the integrity of the cerebral serotonin system. It was shown that brain serotonin binding is associated with for non-motor symptoms, and that preservation of serotonergic functions may be predictive of DBS effects.
Impressive manuscript, I have just a couple of question:
- I think it makes sense to briefly explain the difference between "hot" (emotion-laden) and "cold" (emotion-independent) cognition.
- Is it possible to add information regarding DBS parameters? DBS protocols are also sometimes different, this information can be important.
- LINE 60: “The affective bias in hot cognition describes the relationship between processing of negative and positive emotional input, where a negative bias is the tendency to register negative stimuli more readily”. - Perhaps a reference is needed here.
Minor criticism:
In the manuscript the abbreviation “ADHD” is not explained (of course many people know it is attention deficit hyperactivity disorder, but not everyone)
List of references is quite full and statistical analysis is proper. I am happy to recommend the manuscript for the publication after correction mentioned above.
Author Response
At the manuscript " Hot and Cold Cognitive Disturbances in Parkinson Patients 2 Treated with Dbs-Stn: A Combined Pet and Neuropsychologi-3 Cal Study" by Dr. Louise M Jørgensen et al authors described results of investigation how the brain serotonin system associates to disturbances in cognitive functions and mood in Parkinson’ disease (PD) patients after deep brain stimulation (DBS). To evaluate cognitive function it was used psychological tests. Authors also used positron –emission tomography (PET) to evaluate the integrity of the cerebral serotonin system. It was shown that brain serotonin binding is associated with for non-motor symptoms, and that preservation of serotonergic functions may be predictive of DBS effects.
Impressive manuscript, I have just a couple of question:
- I think it makes sense to briefly explain the difference between "hot" (emotion-laden) and "cold" (emotion-independent) cognition.
We have added the extended explanation:
“Thus, hot cognition refers the interface between emotion and cognition in humans; in other words, the cognitive processing of emotional stimuli, unlike cold cognition that does not include processing of emotional stimuli”.
- Is it possible to add information regarding DBS parameters? DBS protocols are also sometimes different, this information can be important.
The patient’s own stimulation settings for optimal symptom relief was used, and varies between participants in terms of a) active lead contact (position 0 1 2 3), b) pulse-width (µs), c) voltage (V) and d) frequency (Hz). The stimulus was biphasic and the following stimulation parameters were representative for the population: Active contact 0-1, 60-120 µs, 1.5-2.8V and 130 Hz. Reports of all subjects for both hemispheres would make table 1 too busy.
- LINE 60: “The affective bias in hot cognition describes the relationship between processing of negative and positive emotional input, where a negative bias is the tendency to register negative stimuli more readily”. - Perhaps a reference is needed here.
We have now added two references, and adjusted the following reference numbers accordingly.
Elliott R, Zahn R, Deakin JFWW, Anderson IM. Affective cognition and its disruption in mood disorders. Neuropsychopharmacology. 2011;36(1):153-182. doi:10.1038/npp.2010.77
Miskowiak KW, Carvalho AF. ‘Hot’ Cognition in Major Depressive Disorder: A Systematic Review. CNS Neurol Disord - Drug Targets. 2015;13(10):1787-1803.
Minor criticism:
- In the manuscript the abbreviation “ADHD” is not explained (of course many people know it is attention deficit hyperactivity disorder, but not everyone):
We have changed “ADHD” in the manuscript to “attention deficit hyperactive disorder (ADHD)”.
List of references is quite full and statistical analysis is proper. I am happy to recommend the manuscript for the publication after correction mentioned above.
Reviewer 2 Report
- A brief summary (one short paragraph) outlining the aim of the paper, its main contributions and strengths.
“Hot and Cold Cognitive Disturbances in Parkinson Patients Treated with Dbs-Stn: A Combined Pet and Neuropsychological Study”
The aim of this paper to demonstrate the correlation between 5HT1BR binding using PET scanning with the presences of cognitive disturbances in Parkinsonian patients treated using DBS STN. Authors are trying to demonstrate that patients with intact 5-HT system can benefit the most from DBS STN. The final goal will be to find if AZ10419369PET can be useful in the screening of putative candidates for DBS STN. This is an important issue in the field. The motor response to DBS STN has been demonstrated and it is the current practice to recommend DBS STN to patients suffering of the motor complications secondary to levodopa treatment. In the current state of the affairs, mood disturbances and cognitive decline remains challenging and not addressed by deep brain stimulation in STN.
From my point of view, the main strength of the paper is the separate analysis of ‘hot’ and ‘cold’ cognitive disturbances and to show that this two parameters are independently affected by DBS STN. DBS STN has long have a reputation of worsening the cognitive decline. DBS therapy has at best mixed cognitive outcomes across studies, targets, and methodologies. Also, in some studies mood disturbances can be worsen by DBS STN. So trying to identify a subset of patients who can best benefit or at least avoid cognitive decline as a secondary effect of the stimulation is very important and is at the very heart of this research.
- General concept comments
There are several limitations and weakness from my point of view:
- The numbers of patients is low to actually generalize the conclusion
- There is a lack of epidemiological data from the healthy volunteers. The epidemiological data of the patients from the previous paper also has some missing data( DBS side effects, inclusion criteria for DBS surgery)
- I could not find several missing measurements during OFF DBS: Working memory, verbal affective memory and Face recognition (ER bias) does not appear in the article.
- DBS has been turned off only for 45 min. There is always a possible carry on effect that could mask results
- What is the level of LDOPA equivalents during the evaluation in OFF DBS STN? L DOPA could also interfere with the evaluation of Mood states during OFF DBS STN
- I could have included a non DBS PD group to trying to isolate some of DBS effects in mood or cognition.
- There is no sufficient power to detect differences in cognitive measures (either hot or cold) and AZ10419369 PET. The authors acknowledge this, but the conclusion are much too confirmatory. Readers should be aware of the weakness of the analysis in the conclusion of the paper and in the abstract to avoid overreaching conclusions.
- Depression was an exclusion criterion: readers need to know how depression was detected (interviews, tests?)
- Why the authors did not study the reverse action of STN ON?
- Are there any anti-depressant used in any of the groups?
- Table 1: why do not include OFF DBS Data in the Table?
- The term baseline is used in reference to what specifically: Initial state of healthy patients only?
This very interesting paper needs some adjustments to be published.
N TORRES
Author Response
General concept comments
There are several limitations and weakness from my point of view:
- The numbers of patients is low to actually generalize the conclusion
The limitation section says:
“The findings should be interpreted with cautions as the study may not have sufficient power to accurately detect group differences in cognitive measures and correlations to [11C]AZ10419369 PET. Moreover, eight of the thirteen controls entered from other studies with missing data from either one of the cognitive measures or PET scan, resulting in a lower sample size in the control group. While a sample size of thirteen is common for PET studies, this is relatively small for psychological studies, and we suggest these data should be replicated in an independent behavioral study.”
- There is a lack of epidemiological data from the healthy volunteers.
We kindly refer to our reply in question 1.
- The epidemiological data of the patients from the previous paper also has some missing data (DBS side effects, inclusion criteria for DBS surgery)
An exclusion criterion from the study was (i) dysregulated PD, which include side effects such as mood disturbance (hyper/hypomania) and levodopa-induced dyskinesia, and (ii) DBS surgery < 3 months. Patients were selected according to the in/exclusion criteria in the previous paper, and not selected according to specific criteria for DBS-STN surgery. In general, DBS-STN is considered in PD patients with motor symptoms not sufficiently controlled by medicine. PD patients with mood disorders are not eligible for DBS-STN surgery.
- I could not find several missing measurements during OFF DBS: Working memory, verbal affective memory and Face recognition (ER bias) does not appear in the article.
Turning DBS-STN off cause immediate return of the disabling motor symptoms, which make the patient unfit to undergo neuropsychological testing. The stimulator was turned back on immediately after PET scan, when the patient had undertaken a short questionnaire and assessment of motor symptoms. Importantly, the neuropsychological tests are validated in a standardized setting (room), and is not applicable at the hospital scanner facilities.
- DBS has been turned off only for 45 min. There is always a possible carry on effect that could mask results
In the absence of any direct measurements of 5-HT in response to DBS being switched off, after having been constantly switched on for up to several years, we find it is difficult to say anything meaningful about how to weigh such time-variant changes in 5-HT and how long (days or weeks) DBS-STN has to be turned off.
- What is the level of LDOPA equivalents during the evaluation in OFF DBS STN? L DOPA could also interfere with the evaluation of Mood states during OFF DBS STN
We cannot exclude that the change in POMS-V could be ascribed to the dopamine system. We have now added the following text to the manuscript:
“Secondary changes in the dopamine system, when turning DBS off, may possibly also affect vigor.”
The L-DOPA equivalents are given in table 1 of the previous paper [ref 17] and refers to the medication, which is the same in the two conditions. We do of course agree that dopaminergic denervation also plays a role. But the present study deals with the serotonergic system. We cannot make any inferences about the dopaminergic system, since we did not make any measurements of that well-investigated system. We propose that the serotonergic and dopaminergic systems decline in parallel.
- I could have included a non DBS PD group to trying to isolate some of DBS effects in mood or cognition.
We appreciate the reviewer’s idea to include a non-DBS PD group to study the DBS effects in mood or cognition. We have now added the following to the limitation section in the manuscript:
“While a sample size of thirteen is common for PET studies, this is relatively small for psychological studies, and we suggest these data should be replicated in an independent behavioral study, which could also include a non-DBS-STN group to study the effects of DBS-STN on mood and cognition in patients with PD.”
- There is no sufficient power to detect differences in cognitive measures (either hot or cold) and AZ10419369 PET. The authors acknowledge this, but the conclusion are much too confirmatory. Readers should be aware of the weakness of the analysis in the conclusion of the paper and in the abstract to avoid overreaching conclusions.
Cold cognition, as indexed by working memory performance (LNS), was significantly lower in patients as compared to controls, and survived correction for multiple comparisons (pFWER). The association between LNS to 5-HTR binding was also significant after correction of multiple comparisons. We believe that we have been quite cautious in our interpretation and conclusion of the results.
Nevertheless, we have made the following changes to the manuscript (abstract):
“Our study suggest that cerebral 5-HTR binding is associated with non-motor symptoms, and that preservation of serotonergic functions may be predictive of DBS-STN effects”.
- Depression was an exclusion criterion: readers need to know how depression was detected (interviews, tests?)
The manuscript refers to previously reported details [ref 17]. In brief, all participants were scored with the self-reported rating scale of The Major Depression Inventory (MDI). The patients were assessed while on their usual anti-parkinson medication. A score > 20 was considered suggestive of depression. None of the participants met the criteria for depression.
- Why the authors did not study the reverse action of STN ON?
Patients have chronic implants with DBS-STN turned on for up to many years. We cannot speculate on how long DBS-STN has to be turned off before before start of the experiment. Disabling treatment for a prolonged period is not considered feasible in these vulnerable patients. The manuscript says:
“Since our study for logistical reasons only involved turning off DBS-STN, we cannot know if the reverse actions occur when turning DBS-STN on”.
- Are there any anti-depressant used in any of the groups?
None of the participants used anti-depressants
- Table 1: why do not include OFF DBS Data in the Table?
Hot and cold cognition was only assessed when DBS-STN was turned on. We kindly refer to question 4.
- The term baseline is used in reference to what specifically: Initial state of healthy patients only?
We have added the following to the manuscript:
“Baseline measures constitute data obtained before turning DBS-STN off”.
This very interesting paper needs some adjustments to be published.
Reviewer 3 Report
Nice review of cognition and mood non motor symptoms associated to DBS of the STN for PD
Author Response
Reviewer 3: Nice review of cognition and mood non motor symptoms associated to DBS of the STN for PD
Reply: Thank you for the comment.
Reviewer 4 Report
Overall, this is a well articulated paper that attempts to correlate PET markers of non-motor cognitive dysfunction in PD patients with behavioral measures. I only have a few comments.
- Even though PET measures did not show significant difference between healthy and PD patients. it will help the readers appreciate if a figure with exemplary PET scans are showed, particulary with DBS on and off.
- Full demographics including the UPDRS of the patients will help to understand the severity of motor impairment
- Why did the authors use pearsons correlation, rather than spearman which is nonparametric.
- The authors state "L-DOPA equivalents were included as a predictor in our model". Did they find any correlation between dopamine depletion and worsening of non-motor symptoms?
Author Response
Overall, this is a well articulated paper that attempts to correlate PET markers of non-motor cognitive dysfunction in PD patients with behavioral measures. I only have a few comments.
- Even though PET measures did not show significant difference between healthy and PD patients. it will help the readers appreciate if a figure with exemplary PET scans are showed, particulary with DBS on and off.
The PET data has previously reported. The present manuscript relates to cognition and the association to 5-HT1BR binding. As mentioned in the introduction of the manuscript:
“With [11C]AZ10419369 PET, we demonstrated a significant loss of 5HT1BR availability in the frontal and parietal cortex, a deficit which correlated to the degree of motor dysfunction. When DBS-STN was turned off, the brain regions with the best preservation of presynaptic serotonin function, namely the temporal and limbic cortex, responded by releasing serotonin. This suggests that the presynaptic serotonergic terminals in the temporal and limbic cortex are still relatively preserved, whereas the frontoparietal regions are more affected and have lost their serotonin releasing capacity. The temporal and limbic cortex are generally associated with important cognitive functions, such as understanding language, learning, remembering verbal information, facial emotional recognition and processing emotional input[18]”.
- Full demographics including the UPDRS of the patients will help to understand the severity of motor impairment
As stated in the manuscript: “Details of the study population, recruitment, in-and exclusion criteria, demographics and clinical characteristics have previously been reported[17].”
- Why did the authors use pearsons correlation, rather than spearman which is nonparametric.
Thank you for the comment. We correlate ordinate data, and has instead applied Spearman. The manuscript has been changed accordingly:
“LNS and ERbias are anticorrelated at a sub-significant level (Spearman correlation coefficient -0.29, p=0.27).”
- The authors state "L-DOPA equivalents were included as a predictor in our model". Did they find any correlation between dopamine depletion and worsening of non-motor symptoms?
We kindly refer to our reply to Reviewer 2, question 6